# Salt Pretreatment-Mediated Alleviation of Boron Toxicity in Safflower Cultivars: Growth, Boron Accumulation, Photochemical Activities, Antioxidant Defense Response

**DOI:** 10.3390/plants11172316

**Published:** 2022-09-04

**Authors:** Özlem Arslan, Şeküre Çulha Erdal, Yasemin Ekmekçi

**Affiliations:** 1Department of Food Processing, University College of Espiye, University of Giresun, 28600 Giresun, Turkey; 2Faculty of Science, Department of Biology, Hacettepe University, 06800 Ankara, Turkey

**Keywords:** *Carthamus tinctorius* L., chlorophyll a fluorescence transient, morpho-physiological and biochemical traits, NaCl pretreatment, nutrient toxicity

## Abstract

The study aims to elucidate alleviant effects of boron (B) toxicity by salt pretreatment (SP) on growth response, phytoremediation capacity, photosynthesis, and defense mechanisms in two safflower cultivars (*Carthamus tinctorius* L.; Dinçer and Remzibey-05). Eighteen-day-old plants were divided into two groups: SP (75 mM NaCl for 5 days) and/or B treatment (C, 2, 4, 6, and 8 mM B for 10 days). Depending on the applied B toxicity, B concentrations in roots and leaves of both cultivars, necrotic areas of leaves, ion leakage (RLR), and H_2_O_2_ synthesis increased, while shoot and root length as well as biomass, water, chlorophyll a+b, and carotenoid content decreased. In addition, chlorophyll a fluorescence results revealed that every stage of the light reactions of photosynthesis was adversely affected under B toxicity, resulting in decreases in performance indexes (PI_ABS_ and PI_TOT_). However, the cultivars tended to induce the synthesis of anthocyanins and flavonoids and increase the activity of antioxidant enzymes (SOD, POD, APX, and GR) to detoxify reactive oxygen species (ROS) under B toxicity. SP mitigated the negative effects of toxic B on biomass, water and pigment content, membrane integrity, photosynthetic activity, and defense systems. Considering all results, Remzibey-05 was able to better overcome the biochemical and physiological changes that may be caused by B toxicity by more effectively rendering B harmless, although it accumulated more B than Dinçer.

## 1. Introduction

Boron (B), an essential micronutrient for plants, plays an important role in many physiological processes such as cell wall synthesis, membrane stability, root and shoot growth, cell division, lignification, protein synthesis, and nucleic acid and carbohydrate metabolism [1,2]. The optimum concentration range of B in plants is very narrow between limits of deficiency and toxicity. Usually 0.5–2.0 ppm is reported to be the optimum soil B range, while lower and higher values indicate deficiency and toxicity. Critical levels depend on soil type, pH, water status, texture, air humidity and temperature, plant species, and genotype [3]. B toxicity threshold values are directly related to B tolerance levels of plants. For tolerant plants, irrigation water containing 2–4 ppm B can be overcome, while irrigation water containing 0.3 ppm B can cause toxic effects on sensitive plants [4]. The uptake of B by plants can only be in the form of small uncharged boric acid [B(OH)_3_] and borate anions [BO_3_^3−^]. Under physiological conditions of plant cells, more than 98% of the total amount of B is found in free form as boric acid, which can freely pass lipid bilayers and binds to molecules with mono, di-, and poly-hydroxyl groups, such as ribose, apiose, sorbitol, and other polyalcohol [5]. The B content in the soil is usually low, resulting in the inability of plants grown in cultivated areas to supply the amount of B required for their development [6]. B deficiency occurs in plants growing in soils with high rainfall, low organic matter, and high pH [7]. On the other hand, as a result of the evaporation of groundwater with a high B content, B can naturally reach toxic levels by accumulating in the soil. In addition, anthropogenic activities such as mining and processing, use of B-contaminated water for irrigation, fertilization, or irrigation strategies cause B to reach toxic levels in the soil [8]. Globally, B rich soils are found in USA, Australia, Turkey, China, Russia, and Argentina [9,10]. Among these countries, Turkey is the country with the largest B reserves in the world with a ratio of 73% [9]. When present in toxic levels, B limits plant growth and production by causing physiological and biochemical dysfunctions in plants. B toxicity leads deterioration of membrane integrity, inhibition of photosynthesis, degradation of photosynthetic pigments, nutrition imbalances, alteration of antioxidant enzymes, and deposition of lignin and suberin [6,8]. These alterations result in visible symptoms such as chlorosis and/or necrosis, which usually occur on the margins and tips of the mature leaves [8,11]. Moreover, oxidative damage occurs due to the overproduction of reactive oxygen species (ROS) and ROS poses a serious threat to cell functioning by damaging lipids and proteins [12,13]. In order to reduce the adverse effects of ROS, plants may activate scavenging mechanisms, including enzymatic and nonenzymatic antioxidant systems [14].

Boron toxicity, which often occurs in arid or semi-arid areas due to limited leaching, is accompanied by salinity and B accumulates as sodium salts due to its soluble nature, especially in areas with poor drainage [15,16]. In addition, B accumulates in the soil at higher concentrations than salt, as it is removed from the soil more slowly than Na^+^, Cl^−^, and SO_4_^2−^ ions during leaching [4]. This combination of B toxicity and salt stress is referred to as “BorSal” [16]. A wide variety of plants have been investigated in studies focusing on the effects of BorSal treated at different stress levels, including pepper [17], poplar tree [18], ornamental shrubs [19], pistachio [20], and maize [21]. In these studies, although there is no consensus on the reciprocal relationships between simultaneous exposure to salinity and B toxicity, it has been reported that B and salinity generally have an antagonistic effect on plants. On the other hand, the sequential occurrence or increasing dominance of many stressors due to changing climatic conditions has created an important need to investigate the effects of sequential exposure to B and salinity on plants. The only study on this topic deals with the acquisition of B tolerance by salt pretreatment in sunflower [22], and in this study, it was reported that the negative effect of B toxicity on growth, water level, B and pigment content, membrane structure, and photosynthesis were mitigated by salt pretreatment.

Safflower (*Carthamus tinctorius* L.), one of the oldest oilseed plants in the world, has many different uses, such as vegetable oil production, medical, cosmetic, and paint industries and biodiesel production through to its high oleic (omega-9) and linoleic (omega-6) acid content [23,24]. This valuable plant is grown in the Mediterranean region and has been reported to be tolerant to drought and salinity [25,26]. However, in arid and semi-arid regions where salinity is common, safflower also faces B stress. The responses of safflower cultivars at different developmental stages to B toxicity have been the subject of some research [27,28,29]. However, the effects of B toxicity on safflower have been determined for the first time with such detailed morpho-physiological and biochemical parameters. In addition, with the knowledge that the pretreatments reduce the negative effects of more severe stresses encountered subsequently, the question is how salt pretreatment will affect the B toxicity effects on safflower? Analysis of the physiological changes associated with such pretreatment may be helpful to our understanding of the strategies of plants to acquire stress tolerance. Prior to this research, increasing levels of salt concentrations (50, 75, and 100 mM) were applied to safflower cultivars as a preliminary study and the highest salt concentration (75 mM) was chosen as the pretreatment concentration, where the effects of salinity stress were not determined in the cultivars. In this study, two safflower cultivars (Dinçer and Remzibey-05) with different salinity and drought responses were used [25,26]. The purpose of selecting the safflower plant in this study is to detail our knowledge about the stress responses of this plant, which is valuable, industrially and agriculturally. Therefore, gradually increased B concentrations (C, 2, 4, 6, and 8 mM B) were applied to two safflower cultivars (Dinçer and Remzibey-05) with or without salt pretreatment (SP, 75 mM NaCl) to explain the interaction between B toxicity tolerance and SP on growth response, B content, phytoremediation capacity, photosynthesis, and defense mechanisms. The objective of this study was to: (1) understand the tolerance level of safflower cultivars in response to different B toxicity levels; (2) explain the presence of mitigating effects of B toxicity with SP; (3) compare two cultivars in response to tolerance levels; and (4) elucidate possible physiological mechanisms in safflower cultivars under B toxicity conditions with or without SP. 

## 2. Results and Discussion

In nature, plants are exposed to various stress factors simultaneously or sequentially during their life cycle due to the synergistic effect of stresses. Simultaneous or sequential exposure to stresses affects tolerance development and plant survival in different ways. It has been suggested that exposing plants to a certain level of stress with pretreatment may enable that plant to perform better against more severe stresses that it will encounter later, and may increase plant survival under other stresses [22]. Therefore, the morphological, physiological, and biochemical responses of two safflower cultivars were investigated to better understand the mechanisms involved in identifying B tolerance and to define the extent of the ameliorative effect of SP on adverse effects that might be caused by B toxicity.

### 2.1. Plant Growth Response to Toxic B with or without SP

Excess B causes growth and development disorders. The root lengths of the cultivars were more affected by B toxicity than the shoot lengths according to corresponding controls (10–30% and 9–19%, respectively) (Figure 1A,B). The reasons for the negative effects of root development at high B concentrations may be related to the decrease in cell division due to low sugar levels in the root tips, and thus the decrease in root meristem growth [30,31]. The mitigating effect of SP on B toxicity in root length was higher for all SP + B treatments in Dinçer (12–20%) and for SP + 2 and SP + 4 mM B treatments in Remzibey-05 (10% and 17%, respectively) than corresponding B treatments. The positive effect of SP is more evident in the root length in Dinçer. In addition, the fresh weight of shoot and root (14–53% and 11–42%, respectively) and dry weight of shoot and root decreased (34–61% and 24–50%, respectively) at 4 mM and higher B treatments (Figure 1C–F). The reasons for the impairment of plant growth and dry weight yield at a high B concentration may be a deterioration of cell wall thickness, which impedes nutrient uptake, and a reduction in the photosynthesis mechanism, CO_2_ assimilation, and sugar metabolism [32,33]. However, shoot dry weight and root fresh and dry weight were higher for SP + 6 mM B and SP + 8 mM B in Dinçer (40–35%, 22–66%, and 18–38%, respectively), while fresh and dry weights of shoot and root were higher for SP + 4 mM B and higher SP + B in Remzibey-05 (18–46%, 29–50%, 12–44%, and 22–35%, respectively) compared to B treatments. These results showed that the reduction in fresh weight and dry weight of shoot and root caused by B was significantly alleviated by the treatments of SP in the cultivars, and this situation was more pronounced in Remzibey-05. Restriction of plant growth is a general consequence of B toxicity, such as formation of chlorosis and/or necrosis in leaves. Likewise, the necrotic areas that form on leaf tips and margins due to B toxicity are shown in Figure 2A. B tends to accumulate primarily in leaf margins of dicotyledons, and B toxicity symptoms mainly appear in the form of terminal and marginal chlorosis on leaves, followed by necrosis [10]. Necrotic areas increased with B concentrations and the percentage of necrotic area according to the leaf area were mostly at the highest B concentrations in both cultivars (86% and 63% at 8 mM B in Dinçer and Remzibey-05, respectively). The results of the present study are consistent with the findings of Wu et al. [34], who reported that chlorosis at high B content mainly occurs at the leaf tip and then spreads incrementally toward the inner part of the leaf center in trifoliate orange. The formation of necrosis and leaf spread from the tips and margins toward the center are explained by the unequal distribution of B in different leaf sections, which is due to the absorption of B enriched xylem sap during transpiration first into the marginal mesophyll cells [34]. It was also determined that the necrotic areas were remarkably decreased morphologically in SP treatments.

The decline in plant growth and development induced by B toxicity has also been associated with plant water use limitation. Macho-Rivero et al. [35] investigated the molecular basis of reduced water transport under B toxicity and observed suppression of several genes encoding aquaporins in root and shoot, that allow water flow from cell to cell and from root to shoot. Relative water content (RWC) gradually decreased in both cultivars at 4 mM B and higher B treatments (22, 34 and 37% in Dinçer and 16, 27, and 36% in Remzibey-05, respectively), and leaf water potential values also showed similar results (Figure 2B,C). Aquea et al. [31] showed that B toxicity repressed genes encoding water transporters in Arabidopsis roots. Reduction in water availability has also been previously reported under B toxicity in canola [36], maize [37], and tomato [38], as B toxicity can inhibit water flow to aboveground parts by reducing water uptake. However, SP prevented the decrease in water content of cultivars (leaf water potential and RWC). SP mitigated B toxicity, as evidenced by an increase in shoot and root length, fresh and dry weight, leaf water content, and a decrease in necrotic area compared to B treatments. These results suggest that SP plays an important role in mitigating B toxicity-induced damage to safflower. 

### 2.2. Allocation of B in Organs and Phytoremediation Potential of Safflower under B Toxicity with or without SP

Uptake of B from soil solution into roots through a combination of diffusion and transport was mediated by transporters (channels or pumps). The activity of B transporters is tightly regulated, depending on the B concentration in the environment to optimize B uptake and maintain nutrient homeostasis in plant tissues [39,40]. Therefore, B passes through the lipid bilayers of roots mainly by diffusion in the form of B(OH)_3_ with high permeability when the rhizosphere has a high B concentration [33,39]. In the study, an increase in the amount of B in the rhizosphere of the plant root led to an increase in the B accumulation in the plant organs (leaf and root) (Table 1). The gradual increase in B concentration was 10.6- to 18.5-fold and 9.3- to 24.1-fold in the leaves, whereas it was 3- to 7.9-fold and 4.9- to 11.4-fold in the roots of Dinçer and Remzibey-05, respectively. Several reports agree with our results, that B accumulates mainly in the leaves and less in the roots [36,41,42], indicating that distribution of B among plant organs correlates with B concentration and excess B is directed to the leaves for accumulation. In addition, Papadakis et al. [33] reported that visible symptoms appeared on the upper leaves of loquat at high B concentrations. This was because B was transported to the upper leaves via the phloem flow by forming a complex with polyols as sorbitol and mannitol, so that the B content was higher in younger leaves than in older leaves. However, in the current study, higher chlorosis was observed in older leaves under B toxicity than in younger ones, indicating that B was transported through the xylem by transpiration streams in safflower cultivars (Figure 2A and Table 1). In addition, the differences between B accumulation in the organs (leaves and roots) of both cultivars were investigated, and Remzibey-05 had higher B content in leaves and roots than Dinçer for all B treatments compared to the control, with the exception of Dinçer at 2 mM B in leaves. Similar results from increases in B and Na uptake under B toxicity were also obtained in safflower cultivars subjected to salt stress and the accumulation of Na in leaves was higher in Remzibey-05 than in Dinçer [26]. In addition to these, phytoremediation potential of safflower for B was evaluated by calculating translocation factor (TF), bioaccumulation factor (BF), and bioconcentration factor (BCF) from the B contents of safflower cultivars in leaves and roots (Table 1). BF and BCF indicate the efficiency of element accumulation in plant tissues, while TF reflects the ability of the plant to transport the element from roots to aboveground parts [15,43]. The TF, BF, and BCF values of the cultivars were higher than 1, and BF > TF > BCF in all B treatments indicated that cultivars took up B from the rhizosphere with the roots and transferred it to the aerial parts, especially the leaves. It was reported that the BF, BCF, and/or TF values exceeded 1 in high B concentrations in *Puccinellia distans* [43], *Puccinellia tenuiflora* [44], poplar [45], and sunflower [22]. Usman et al. [46] reported that the plant remediated the soil with respect to the element that had high/toxic concentrations through phytoextraction capacity when BF, BCF, and TF were greater than 1. Additionally, the study by Chen et al. [45] found that Populus, a woody plant with rapid biomass production, possessed BCF and TF > 1 values, and the results were similar when compared with the B tolerant plant *Puccinellia distants* [43]. Although the production of root and shoot biomass decreased with the increase in B concentrations of both safflower cultivars; the values of BF, BCF, and TF were above 1, indicating that safflower has a high B phytoextraction potential (Figure 1 and Table 1).

On the other hand, the SP treatments reduced the accumulation of B in the tissue of safflower cultivars, and the B contents were 47–75% and 57–74% lower in the leaves, and 20–31% and 40–51% lower in the roots of Dinçer and Remzibey-05, respectively, in SP treatments compared with B treatments. Salt stress may lead to stomatal closure by reducing the osmotic potential in the soil, which may have limited the uptake of B from the roots and its transport to shoot with reduced evapotranspiration [17,18]. In addition, salt stress may have hindered B uptake by regulating the gene expression of aquaporin, which allows the exchange of B via the plasma membrane at toxic B concentrations [15,16]. The increase in growth and water content of salt pretreated safflower cultivars under toxic B conditions may indicate that the decrease in expression of aquaporin, together with the decrease in evaporation, has a predominant effect in limiting B uptake in the root and transport to the shoot. Additionally, B transporters such as AtBOR4 and HvBOR2 are mainly responsible for tolerance to excess B by exclusion from tissues [39,47,48]. SP may have decreased the synthesis of aquaporin while increasing the expression of the transporter that enables B exclusion in safflower cultivars under toxic B conditions. Moreover, Dinçer had higher B content than Remzibey-05 in all SP treatments, and B uptake in roots and B transport in leaves were also lower in Remzibey-05 according to B concentrations, except for SP + 4 mM B compared with 4 mM B. In addition, the values of TF, BF, and BCF were higher than 1 in SP treatments, but these values were lower than B treatments, indicating that the SP reduced B uptake and B transfer into leaves of both cultivars, but the phytoextraction potential was still high.

### 2.3. Chlorophyll a Fluorescence (ChlF) Transient Analysis in Safflower Leaves under B Toxicity with or without SP

Excess B in the growth medium leads to morphological changes as well as impaired metabolic functions in plants. Although the physiological/biochemical basis of B toxicity has not been fully elucidated, reduced plant growth and visible symptoms in leaves can be attributed to the major metabolic effects of B. These include: (1) alterations in the cell wall structure and matrix stability; (2) deterioration of primary metabolism by binding to ribose in ATP, NADH, or NADPH; (3) impairment of cell division and expansion by binding to ribose, both as simple polyol and as an RNA component; and (4) reduction of cytosolic pH, affecting protein structure and synthesis [5,49]. In this regard, photosynthesis is one of the most important metabolic processes disrupted by toxic B. Many recent studies have shown that the reduction of photosynthesis in plants treated with toxic B is due to non-stomatal and/or stomatal limitations [41,49,50,51,52]. It was reported that B toxicity significantly reduces the transpiration rate due to activation of ABA metabolism, maintaining the water status in Arabidopsis [35]. Thus, stomatal constraints reduce water loss while limiting CO_2_ inflow. The significant decrease in water content under B toxicity in safflower cultivars suggests that non-stomatal limitations (biochemical limitations) are more effective in reducing photosynthesis (Figure 2). In this study, biochemical limitations of photosynthesis in safflower cultivars were analyzed using the polyphasic chlorophyll a fluorescence (ChlF) kinetics technique, which is widely used to show the change in function, conformation and structure of the photosynthetic apparatus under different conditions in photosynthetic organisms [53,54,55]. The OJIP transient is presented in Figure 3 as kinetics of relative variable fluorescence at any time V_t_ = (F_t_ − F_0_)/(F_m_ − F_0_) and as differences of normalized all stress and pretreatment groups transient minus the C transient (ΔV_t_). The fluorescence curves of C groups showed a typical OJIP shape, while the increase of the initial ChlF level (O step), the decrease of the maximal level (P step), and the alteration in the fluorescence curves after exposure to B and SP + B treatments are visible in safflower cultivars (Figure 3A,B). Analysis of the shape of the OJIP curve showed that B treated plants had a higher fluorescence increase in the O-J phase and a slower fluorescence increase in the J-P phase compared with SP + B treated plants, and these differences between SP + B and B treatments increased with B concentration. Moreover, no specific differences were observed between the polyphasic ChlF induction curves of the cultivars (Figure 3A,B). Therefore, the transitions of the double-normalized fluorescence curves generated by subtracting the normalized fluorescence values (between O and P steps) are shown in Figure 3C,D to reveal the differences between cultivars and to better analyze the change in the shape of the induction curve. Significant changes in the shape of the curve were observed in both safflower at all stages of the treatments, although this change was more pronounced in Dinçer. In the O–I phase, the drawn curves had higher fluorescence intensity as positive deviation than the C groups, while in the I–P phase, the curves had lower fluorescence intensity as negative deviation. It was also determined that the increase in fluorescence intensity at 4 mM and higher B treatments were higher than that at the corresponding SP + B treatments, indicating that the damage caused by B toxicity was avoided by the SP treatments. In addition, the difference curves were plotted separately to show the bands hidden between the O, J, I, and P steps in these fluorescence induction curves (Figure 4). The ΔW_OK_ and ΔW_OJ_ peaks revealing the L- and K-bands, respectively, appeared in all B treatments in both cultivars, except for 2 mM B in the ΔW_OK_ curve in Dinçer (Figure 4A–D). The presence of the L-band provides information about the energetic connectivity and grouping probability of PSII units as well as the utilization of excitation energy, while the K-band indicates the balanced electron transfer from the oxygen evolving complex (OEC) to P680^+^ and subsequently to Q_A_^−^ [53,56]. The extent of the L-band increased in Dinçer depending on B concentrations, while it remained the same in Remzibey-05 at 4 mM and higher B treatments (Figure 4A,B), implying that energetic connectivity losses occurred between the reaction centers and their antennae complexes, resulting in weaker utilization of excitation energy and lower stability of PSII units under B toxicity [53,57]. The K-band, which occurs within the 200–300 µs range of the ChlF induction curve, increased its extent with B concentration in the cultivars, except at 8 mM B (Figure 4C,D). Under stressful conditions, a positive K-band indicates inactivation of OEC due to damage to the Mn-complex, as well as downregulation of genes encoding PsbC and PsbE involved in binding with PsbO, leading to disruption of electron transfer from the OEC to the reaction center of PSII [56,57]. Our results were consistent with previous studies which showed that B toxicity resulted in impaired energy transfer from the light-harvesting complex and the OEC to the reaction center of PSII, which could lead to an imbalance between the donor and acceptor sides of PSII, resulting in Q_A_^−^ accumulation [22,52]. Therefore, the restricted re-oxidation of Q_A_^−^ was evidenced by the visible J-band, and the extent of the J-band increased depending on the B concentrations in cultivars, with this increase being more pronounced in Dinçer (Figure 4E,F). The J-band provides information about the Q_A_^−^ reduction and re-oxidation rates, and the increase in the extent of the J-band indicates that B toxicity leads to accumulation of Q_A_^−^ and inhibition of Q_A_^−^ re-oxidation [58]. On the other hand, the amplitude of the ΔW_OK_ and ΔW_OJ_ curves was lower than the control level in both cultivars under the SP treatments. Moreover, a low extent of the negative L- and K-band was observed in Dinçer and Remzibey-05, with the exception of SP + 8 mM B. The negative L-band indicates a more efficient use of excitation energy and higher stability due to better connectivity between PSII units, while the negative K-band indicates that the stability of the OEC is maintained [53]. The amplitude of the ΔW_OI_ curve was also lower in the SP treatments, but the J-band, which was still present in salt pretreatments, was more pronounced in Dinçer. Moreover, the presence of the G-band in the safflower cultivars was found to be a result of the normalization of the I-P phase (ΔW_IP_), which is related to electron transfer from PSII to PSI (Figure 4G,H). The G-band signals the state of the protonated secondary quinone acceptor (Q_B_H_2_) during electron transport through PSI [59]. The size of the G-band was more increased in B and SP treatments in Remzibey-05, and this result showed that Dinçer could better cope with the disruption of the protonated secondary quinone acceptor under B toxicity and better avoid it with SP treatments. Consequently, the K-, L-, J-, and G- bands showed that B toxicity affected the light reactions of photosynthesis in safflower cultivars, but the damage to the donor part of PSII could be more effectively prevented by SP treatment than that to the acceptor part.

In this study, we also calculated performance indexes (PIs) and their components, which are a robust mathematical expression of the changes in polyphasic chlorophyll fluorescence curves and OJIP steps and are shown in Figure 5. PIs are a multiparametric expression of successive steps in the primary photochemical reactions from the absorption of photons by PSII reaction centers to the reduction of intersystem electron transport (PI_ABS_) or reduction of PSI end electron acceptor (PI_TOT_) [53,56]. PIs decreased by more than 75% compared to controls at 4 mM and higher B concentrations in safflower cultivars. The reduction in PIs was associated with changes in their components, and all components were significantly reduced by B toxicity. RC/ABS, which is an indicator of efficiency expressed as the concentration of reaction centers in the total pool of chlorophylls [58], decreased with increasing B concentration in Dinçer (9–69%) and Remzibey-05 (17–52%). The decrease in RC/ABS may be due to a decrease in the efficiency of the antenna size or may be due to the conversion of reaction centers from active to the inactive state, which may not decrease Q_A_^−^ [53]. 

The reduction of RC/ABS was accompanied by the trapping of excitation energy in PSII [φ_P0_/(1 − φ_P0_)], electron transport from Q_A_ to Q_B_ [ψ_0_/(1 − ψ_0_)], and from Q_B_ to PSI acceptors [δ_R0_/(1 − δ_R0_)] (Figure 5). Moreover, φ_P0_/(1 − φ_P0_) was the least affected (29–41% in Dinçer and 18–38% in Remzibey-05), whereas ψ_0_/(1 − ψ_0_) decreased the most among these parameters (8–91% in Dinçer and 20–79% in Remzibey-05). Therefore, B toxicity caused a gradual decrease in energy transduction from the absorption of photons, followed by the formation and transport of electrons through PSII in safflower cultivars. As a result, the electron transfer from Q_B_ slightly increased under B toxicity, which can be associated with the formation of cyclic electron flow around PSI. PIs and their components were significantly and markedly increased under SP treatments compared to B toxicity. These results indicate that the SP treatments can prevent the damage caused by B toxicity to the light reactions of photosynthesis, and the prevention was more pronounced in Remzibey-05 than Dinçer. In addition, PI_TOT_ decreased more than PI_ABS_ in B and SP + B treatments. The reason is that PI_TOT_, a powerful parameter describing the transition potential of electrons from exciton to reduction of PSI end acceptors, better reveals the effects of abiotic stress on light reactions [60,61,62].

### 2.4. Alteration of Biochemical Stress Indicators under Toxic B with or without SP

Chlorosis occurs depending on the structural and compositional changes caused by B toxicity in the leaf tip and center by affecting the cell wall, chloroplast, and plastoglobulus. Chloroplast degradation affects both photosynthesis and pigment synthesis and structure of leaves [34]. Photosynthetic activity was affected by B toxicity in safflower cultivars (Figure 3, Figure 4 and Figure 5). In addition, B treatments significantly decreased the photosynthetic pigments of the cultivars compared to the control (Table 2), and the increase in necrotic areas on leaves is indicative of the decrease in photosynthetic pigments (Figure 2). The total chlorophyll (Chl a+b) content gradually decreased in cultivars for 4 mM B and higher B treatments and the decrease was higher in Dinçer (18, 54, and 72%, respectively) than Remzibey-05 (17, 34, and 48%, respectively). The decrease in Chl a+b content under B stress may be due to inhibition of chlorophyll and protochlorophyllide reductase biosynthesis and/or increased synthesis of degradative enzymes such as δ-aminolevulinic acid and protochlorophyllide [63,64]. Safflower cultivars were able to overcome the negative effects of B toxicity on Chl a+b content with SP treatments, and Remzibey-05, in particular, was able to maintain Chl a+b content at the control level with SP treatments, except SP + 8 mM B. Moreover, the decrease and increase of light-harvesting molecules (Chl a+b) triggered the increase and decrease, respectively, of the extent of the L-band in photosynthesis activity of safflower cultivars under B and SP + B treatments (Figure 4A). The decrease in Chl a+b under high B toxicity was accompanied by a decrease in carotenoids, and the carotenoid content decreased in Dinçer at 6 and 8 mM B treatments (37% and 63%, respectively), whereas it decreased by 80% in Remzibey-05 only at the highest B treatments. Carotenoids protect the photosynthetic apparatus from photooxidation by dissipating energy; they also act as alternative antennae to capture and absorb light and transfer energy to chlorophylls [65]. Our results are consistent with the results of the study by Navaz et al. [66], where it was reported that the decline in Chl a+b and carotenoid contents under B stress may be associated with H_2_O_2_ accumulation, which damages the photosynthetic apparatus (Figure 6). However, the change in carotenoid content was not significant in safflower cultivars under SP + B compared with B treatments (Table 2). The fact that the changes in carotenoid contents at SP + B were statistically insignificant compared to high B toxicity may be related to the increase in the amount of Chl a+b pigments that were sufficient for light absorption and transfer energy to reaction centers (RC) in SP treatments.

Anthocyanins belong to the group of water-soluble non-photosynthetic pigments in plants and are synthesized as an end product of the flavonoid synthetic pathway [67]. Anthocyanins and flavonoids act as antioxidants and protect photosynthetic mechanisms from oxidative damages [68,69]. Anthocyanin content was markedly increased in 4 mM B and higher B treatments in Dinçer (7.2- and 9-fold), and 2 and 4 mM B treatments (2.1- and 2.6-fold, respectively) in Remzibey-05 compared to controls (Table 2). However, the SP treatments significantly increased anthocyanin content in both cultivars compared to B treatments (1.22- and 1.66- fold in Dinçer, and 1.38- and 4.27-fold in Remzibey-05). Similar to anthocyanin content, the flavonoid content of cultivars also increased significantly in B and SP + B treatments (Table 2). Similarly, it was reported that flavonoid and anthocyanin contents increased in *Solanum lycopersicum* and *Arabidopsis thaliana* under B toxicity [14,65]. Anthocyanins and flavonoids, as vital secondary metabolites, act as both antioxidants and chelating agents for metals and metalloids in plants [68]. In the presence of excessive B stress, the formation of B-anthocyanin complexes in vacuoles by anthocyanins reduces cellular free B concentrations as well as the potential adverse effects of toxic B [5]. On the other hand, flavonoids also alter the lipid packaging arrangement by modifying peroxidation kinetics, thereby reducing membrane fluidity, which limits the passage of B through the membranes [70]. The increase in anthocyanin and flavonoid contents at SP treatments indicates that these molecules form a complex with B and/or reduce the passage of B through biological membranes, rendering B harmless.

Ion leakage and hydrogen peroxide (H_2_O_2_) content have been used as indicators of membrane damage and oxidative stress caused by various environmental stresses. Increased production of ROS due to stress leads to lipid peroxidation through the oxidation of unsaturated fatty acids, resulting in rupture of biological membranes and thus increased permeability, which in turn results in leakage of cell contents and ions [51,71]. In addition, reduced ChlF efficiencies cause photoinhibition in leaves and overproduction of ROS that cause peroxidation of membrane lipids, denaturation and aggregation of proteins, DNA fractures, and inactivation of enzymes [11]. The relative leakage ratio (RLR) increased 2.05- to 2.87-fold in Dinçer and 3.18- to 4.15-fold in Remzibey-05 at 4 mM B and higher B concentrations (Figure 6A). Similar results were obtained for H_2_O_2_, and H_2_O_2_ accumulation increased by 1.58- to 2.66-fold at 4 mM B and higher B concentrations in Dinçer and by 50% and 56% at 6 and 8 mM B in Remzibey-05, respectively (Figure 6B). On the other hand, the fact that RLR levels in safflower cultivars increased more than H_2_O_2_ accumulation indicated that the enhancement of RLR may be related to other ROS. It has been reported that the accumulation of hydroxyl radicals, which are called free radicals and are a more potent oxidant than H_2_O_2_, increases due to B toxicity, leading to cellular membranes degradation [11,72]. B toxicity inhibits the transport of electrons and result in the production and accumulation of H_2_O_2_ in safflower. However, SP prevented the increase of ion leakage in the cultivars and kept it close to the control values (Figure 6A). The ability to maintain stable ion leakage values indicates that the structural and functional integrity of the cellular membranes was preserved by salt pretreatments. Meanwhile, the H_2_O_2_ content increased by 98% and 116% in Dinçer and 43% and 33% in Remzibey-05 for SP+6 and SP + 8 mM B treatments, respectively, compared with the control. Accordingly, the maintenance of membrane integrity in SP indicates that H_2_O_2_ functions as a signaling molecule rather than a reducing or oxidizing agent in safflower cultivars.

Plants activate an antioxidant defense mechanism to maintain cell membranes and prevent excessive ROS production, improving plant tolerance to oxidative stress. In this study, the effect of toxic B and SP treatments on antioxidant enzymes [superoxide dismutase (SOD), peroxidase (POD), ascorbate peroxidase (APX), and glutathione reductase (GR)] in safflower cultivars was also investigated. These antioxidants play a critical role in the alleviation of abiotic stresses. Among them, SOD catalyzes the first step that converts O_2_^−^ to H_2_O_2_ [51,72]. In particular, the activities of SOD increased 61- to 413-fold at 4 mM B and higher B concentrations in Dinçer and 149- and 180-fold at 6 and 8 mM B, respectively, in Remzibey-05 (Figure 6C). Upregulation of SOD activities has been reported in several plants under B toxicity, e.g., tomato [7], beet [73], and watermelon [63]. Although the activity of SOD is lower under SP treatments than B toxicity, it is significantly higher compared to the control. This is related to a lower release of free ROS due to a lower B uptake at SP. The activities of other antioxidant enzymes (POD, APX, and GR) that play a role in detoxification of ROS and regulation of H_2_O_2_ production at the intracellular level also changed at different levels in safflower cultivars depending on the treatment. In Dinçer, POD activity at 2 mM B (37%), APX activity at 2 and 4 mM B (3.26- and 4.97-fold, respectively), and GR activity at 4 mM B (3.14-fold) increased significantly (Figure 6D–F). Moreover, only the activities of APX and GR increased 2.06-fold at 4 mM B in Remzibey-05 (Figure 6E,F). According to POD, APX, and GR enzyme activity results and H_2_O_2_ accumulation, it might be suggested that antioxidant enzymes play a role in ROS detoxification in safflower cultivars at 2 and 4 mM B treatments. Our results confirm previous studies that found an increase in the activities of POD, APX, and/or GR under B toxicity [1,2,11,38]. On the other hand, POD, APX, and GR activities increased significantly at SP + 2, SP + 4, and/or SP + 6 mM B in Dinçer, while only the activities of APX and GR increased at SP + 4 and SP + 6 mM B in Remzibey-05 compared to the control and B toxicity (Figure 6D–F). These results indicate that POD could not be sufficiently activated to overcome the H_2_O_2_ accumulation of Remzibey-05 (Figure 6B,D–F). However, the SP treatments showed that the AsA-GSH cycle functioned more effectively in H_2_O_2_ detoxification in both safflower cultivars, except for SP + 8 mM B. The activities of antioxidant enzymes results demonstrate that SP can alleviate B toxicity, except the severest B concentration, by modulating redox balance through the activation of the antioxidant responses.

## 3. Materials and Methods

### 3.1. Plant Materials, Growth, and Treatment Conditions

Safflower (*Carthamus tinctorius* L.) seeds of the cultivars (Dinçer and Remzibey-05) were obtained from the Central Research Institute of Field Crops in Turkey. Seeds were surface sterilized [5% sodium hypochlorite (NaOCl) solution for 3 min] and imbibed in distillated water for 2 h. After incubation, 8 seeds were sown in plastic pots (14 cm diameter and 13 cm height) filled with perlite and thinned to 5 seedlings after emergence. The pots were watered every other day with half-strength Hoagland’s solution [74]. Plants were grown in a controlled growth chamber, with a temperature of 25 ± 1 °C, a 16 h photoperiod, a relative humidity of 60 ± 5%, and an irradiance 250 μmol m^−2^ s^−1^. The concentration of SP was determined as 75 mM based on preliminary results. Plants were grown for 18 days under these conditions and then randomly divided into groups treated with salt (75 mM NaCl) for 5 days and/or with B (2, 4, 6, and 8 mM H_3_BO_3_) for 10 days (SP—salt pretreatment/B—boron stress/SP + B—salt pretreated boron treatments). The experimental design is detailed in Figure 1. 

### 3.2. Growth Parameters, Necrosis Area, and Water Content of Leaves

Shoot and root lengths of safflower seedlings were measured (mm plant^−1^). Three plants from each group were taken randomly to determine fresh weight (g plant^−1^) and then kept in an oven at 80 °C for 48 h to determine shoot and root dry weight (g plant^−1^). Images of the leaf samples were taken and analyzed using the ImageJ@ program to determine the necrosis area (%). The water status of leaves [2 leaf discs (R = 0.5 cm in the middle of the leaf) of each treatment and 6 replicates] was evaluated by calculating RWC as: RWC (%) = [(FW − DW)/(SW − DW)] × 100, where FW is the fresh weight, DW is the dry weight, and SW is the water-saturated weight [75]. Leaf water potential (6 replicates) was measured directly from the WP4 Dewpoint Potential Meter (WP4-T/Operator’s Manual Version 2.2, Decagon Devices, Inc., Pullman, WA, USA).

### 3.3. B Contents and Pigment Analysis

Harvested seedlings were washed three times in deionized water, then leaf and root tissues were collected separately (6 replicates) and dried at 80 °C for 48 h. Next, 0.2 g of dried tissues were ground to a powder and burned in a muffle furnace at 550 °C for 5 h. The residue was brought to a standard volume with 1 M HNO_3_ and then filtered through Whatman paper. The B content (mg kg^−1^ DW) in the tissues was quantified by inductively-coupled plasma-atomic emission spectroscopy analysis (ICP-AES, IRIS Intrepid, Thermo Elemental, Waltham, MA, USA). Subsequently, the B contents of leaves and roots were calculated (mg kg^−1^ DW) (6 replicates) and B translocation factor [TF, B concentration in leaf (mg kg^−1^ DW)/B concentration in root (mg kg^−1^ DW)], bioaccumulation factor [BF, B concentration in leaf (mg kg^−1^ DW)/B concentration in soil (mg kg^−1^ DW)], and bioconcentration factor [BCF, B concentration in root (mg kg^−1^ DW)/B concentration in soil (mg kg^−1^ DW)] were calculated (6 replicates) from the obtained data according to Yoon et al. [76] and Roccotiello et al. [77]. For each treatment, photosynthetic pigments [chlorophyll (a + b) and carotenoids (x + c)] were extracted from leaf discs (R = 0.6 cm and 6 replicates in 100% acetone and the absorbance of the extracts was measured at 470, 644.8, and 661.6 nm. The contents of photosynthetic pigments (mg cm^−2^) were calculated using adjusted extinction coefficients [78]. Anthocyanin content was determined from acidified methanol [1 mL of methanol:water:HCl (79:20:1)] extractions of the leaf samples (6 replicates) and the absorbance was measured at 530 and 657 nm. The anthocyanin content was calculated according to Mancinelli et al. [79] and expressed as mg g^−1^ FW. Flavonoid content of leaves (0.1 g of fresh leaf tissue and 6 replicates) was determined according to the method of Mirecki and Teramura [80]. The leaf samples were extracted in acidified methanol [6 mL of methanol:water:HCl (79:20:1)]. The relative flavonoid content (A_300_) was estimated from the absorbance at 300 nm of the acidified methanol leaf extracts and calculated as the percentage of the control plants (C).

### 3.4. Polyphasic ChlF Measurement

Polyphasic OJIP fluorescence transients were performed on selected leaves (6 replicates) of the cultivars using a Handy PEA (Plant Efficiency Analyser, Hansatech Instruments Ltd., Norfolk, UK) fluorimeter. After a 30-min dark adaptation, the measurement consisted of a single strong 1 s light pulse [650 nm peak wavelength; 3000 μmol (photon) m^−2^ s^−1^ an excitation intensity sufficient to ensure the closure of all PSII reaction centers] provided by three LEDs. Fluorescence intensity at 20 µs (F_0_), 100 µs, 300 µs (F_K_), 2 ms (F_J_), 30 ms (F_I_), and maximum fluorescence (F_P_) were recorded. The recorded data were analyzed using BiolyzerHP3 to detect the physiological state of the safflower plants. Relative variable fluorescence [ΔV_t_: between the steps O and P, ΔV_t_ = [(F_t_ − F_0_)/(F_P_ − F_0_)]_(treatment)_ − [(F_t_ − F_0_)/(F_P_ − F_0_)]_(control)_] was calculated to determine the differences between safflower cultivars in response to toxic B with or without SP. To further illustrate the differences between cultivars in response to the treatments, relative fluorescence between the steps O and K [20 and 300 μs, respectively = V_OK_ = (F_t_ − F_0_)/(F_K_ − F_0_)], O and J [20 μs and 2 ms, respectively = V_OJ_ = (F_t_ − F_0_)/(F_J_ − F_0_)], O and I [20 μs and 30 ms, respectively = V_OI_ = (F_t_ − F_O_)/(F_I_ − F_O_)], and I and P [30 ms and at the peak P of OJIP, respectively = V_IP_ = (F_t_ − F_I_)/(F_P_ − F_I_)] were normalized and presented as the kinetic difference ΔV_OK_ = V_OK(treatment)_ − V_OK(control)_, ΔV_OJ_ = V_OJ(treatment)_ − V_OJ(control)_, ΔV_OI_ = V_OI(treatment)_ − V_OI(control)_ and ΔV_IP_ = V_IP(treatment)_ − V_IP(control)_, respectively [53,60]. In addition, the performance indexes (PI_ABS_ and PI_TOT_) were calculated from the components to determine the difference between the cultivars [PI_ABS_: (RC/ABS) − [φ_P0_/(1 φ_P0_)] [ψ_0_/(1 − ψ_0_)], performance index (potential) for energy conservation from photons absorbed by PSII to the reduction of intersystem electron acceptors; PI_TOT_: PI_ABS_ [(δ_R0_/(1 − δ_R0_)], performance index (potential) for energy conservation from photons absorbed by PSII to the reduction of PSI end acceptors; RC/ABS: Q_A_ reducing RCs per PSII antenna chlorophyll; φ_P0_/(1 − φ_P0_): efficiency of primary photochemistry trapping, ψ_0_/(1 − ψ_0_): the ratio of electrons removed from the system and electrons accumulated in the system; and δ_R0_/(1 − δ_R0_): the efficiency of intersystem electron transport to PSI end electron acceptors] [53,81].

### 3.5. Relative Leakage Ratio, H_2_O_2_ Content, and Antioxidant Enzyme Activities

Relative leakage ratio (RLR) was calculated indirectly as the leakage of UV-absorbing substances using the protocol described by Redmann et al. [82] with some modifications. Five leaf discs (R = 0.6 cm and 3 replicates) were cut and shaken for 24 h in 10 mL of distilled water. Then, the leaf discs were placed in liquid nitrogen and the absorbance values of the incubation solutions were recorded at 280 nm (A_280_). The leaf discs were placed back to the tubes and shaken for an additional 24 h. The absorbance of the incubation solutions was again determined at 280 nm (A’_280_). The RLR was calculated according to the A_280_/A’_280_ formula. H_2_O_2_ content (µmol g^−1^ FW^−1^) was determined according to the method of Esterbauer and Cheeseman [83]. Fresh leaf tissue (0.1 g and 3 replicates) was homogenized in 0.1% trichloroacetic acid at 4 °C and centrifuged at 10,000 rpm for 15 min. To determine the H_2_O_2_ content, 0.1 M Tris-HCl (pH 7.6) and potassium iodide reagent were added to the supernatant. The reaction mixture was kept in the dark for 90 min and the absorbance was read at 390 nm. The H_2_O_2_ content was calculated according to the standard curve. To determine the enzyme activities, fresh leaf samples (0.5 g and 3 replicates) were ground with liquid nitrogen and the soluble protein was extracted in respective extraction buffer. Protein concentrations from leaf extracts were determined according to Bradford [84]. Fine powder was homogenized in 1 mL of buffer containing 9 mM Tris–HCl buffer (pH 6.8) and 13.6% glycerol and total activity of SOD (EC 1.15.1.1) was determined as described by Beyer and Fridovich [85]. One unit of SOD activity was defined as the amount of enzyme required to cause 50% inhibition of NBT photoreduction and expressed as U g protein^−1^ APX (EC 1.11.1.11) activity was assayed according to the method of Wang et al. [86] and the buffer in which the fine powder of leaf tissues were homogenized contained 50 mM Tris–HCl (pH 7.2) buffer, 2% PVP, 1 mM Na_2_EDTA, and 2 mM ascorbate The enzyme activity was calculated from the initial rate of the reaction using the extinction coefficient, ε, of ascorbate (ε = 2.8 mM cm^−1^) at 290 nm and expressed as µmol ascorbate min^−1^ mg protein^−1^ The homogenization buffer of the leaves extracted for GR (EC 1.6.4.2) and guaiacol POD (EC 1.11.1.7) contained 100 mM potassium phosphate buffer (pH 7.0), 2% PVP, and 1 mM Na_2_EDTA. GR activities were determined according to Rao et al. [87] and calculated from the initial rate of the reaction after subtracting the non-enzymatic initial oxidation rate using the extinction coefficient of NADPH (ε = 6.2 mM cm^−1^) at 340 nm. GR activities were expressed as µmol NADPH min^−1^ mg protein^−1^. Guaiacol POD activity was based on the determination of guaiacol oxidation (ε = 26.6 mM cm^−1^) at 470 nm by H_2_O_2_ [88]. A unit of peroxidase activity was defined as µmol H_2_O_2_ decomposed per minute per milligram of protein.

### 3.6. Statistical Data Analysis

Experiments were performed in a completely randomized design by three replicates and 300 plants in 60 pots. Experiments were performed with 3–6 replicates (one plant from different pots per replicate). To confirm the variability of data and validity of results, all the data were subjected to analysis of variance (*ANOVA*) and differences between cultivars and treatments were calculated according to the least significant difference (LSD) test at a 95% probability level and significant differences (*p* < 0.05) within each group were indicated by different letters (except plotted curves). All the analyses were performed using the *SPSS* v *20.0* (Chicago, IL, USA).

## 4. Conclusions

B toxicity and salinity are important stress factors threatening arid and semi-arid agricultural regions in changing environmental conditions. Although the effect of coexistence exposure on plant growth and development has been studied in detail, the effect of sequential exposure on morphological, physiological, and molecular pathways remains to be clarified. On this basis, this study demonstrated that safflower cultivars that perceive the presence of stress by being exposed to low NaCl concentrations can mitigate the negative effects that may result from subsequent B toxicity. While B uptake increased in safflower cultivars, shoot and root fresh and dry weights decreased due to B toxicity, especially in root. In addition, B toxicity caused the formation of necrosis by triggering a decrease in water content, loss of pigment, and a decrease in photosynthetic activity along with an increase in ion leakage and H_2_O_2_ content in leaves. On the other hand, cultivars sought to activate some antioxidant enzymes to detoxify stress-induced ROS. SP attenuated the detrimental effects of B by reducing B uptake into roots, reducing B entry from the cell wall, and chelating intracellular B. SP ensured the preservation of water content and morphological structure of safflower cultivars. In addition, polyphasic ChlF kinetics was used to reveal changes in the bioenergetic state of the photosynthetic apparatus in safflower cultivars. Photochemical (energy absorption, dissipation, and trapping) and thermal (electron transport from PSII reaction center to PSI electron acceptors) deteriorations and membrane structure disruption caused by B toxicity photosynthesis mechanisms were alleviated by SP and the continuity of photosynthetic activity was ensured. In addition, SP has been shown to be more effective in promoting defense mechanisms in safflower cultivars. Considering all the results, Remzibey-05 was more successful than Dinçer in eliminating the detrimental effects of B, although it accumulated more B, and SP was quite effective in reducing the effects of B toxicity. One of the important findings of our study was that safflower cultivars accumulate in small quantities of B in the roots and a considerable amount of B is transported to the shoots and accumulates in the leaves. Accordingly, it has been suggested that safflower has high phytoextraction potential, and that this plant can be used to decontaminate B contaminated soil. After that, the plants obtained from contaminated soil can be used industrially. In fact, with more detailed studies, it should be determined whether the safflower planted in soils with B excess can be used for agricultural purposes by grading B accumulation in every part of the plant, especially in the seeds. This study provides a preliminary report on the effects of B toxicity with SP on leaf physiology and biochemistry and phytoextraction capacity in safflower cultivars.

## Data Availability

All data are contained within the article.

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
