# Peer review of "Salt Pretreatment-Mediated Alleviation of Boron Toxicity in Safflower Cultivars: Growth, Boron Accumulation, Photochemical Activities, Antioxidant Defense Response"

_plants, 2022, doi:10.3390/plants11172316_

Round 1

Reviewer 1 Report

 English should improve by a native person. The paper suffers from a poor English structure throughout and cannot be published or reviewed properly in the current format. The manuscript requires a thorough proofread by a native person whose first language is English. The instances of the problem are numerous and this reviewer cannot individually mention them. It is the responsibility of the author(s) to present their work in an acceptable format. Unless the paper is in a reasonable format, it should not have been submitted.

2.    The novelty of the study needs to be highlighted compare to other similar studies.

3.    Discussion is weak. The discussion needs enhancement with real explanations not only agreements and disagreements. Authors should improve it by the demonstration of biochemical/physiological causes of obtained results. Instead of just justifying results, results should be interpreted, explained to appropriately elaborate inferences. Discussion seems to be poor, didn't give good explanations of the results obtained. I think that it must be really improved. Where possible please discuss potential mechanisms behind your observations. You should also expand the links with prior publications in the area, but try to be careful to not over-reach. For the latter, you should highlight potential areas of future study.

4.    The scientific background of the topic is poor. In "Introduction" and "Discussion", the authors should cite recent references between 2016-2020 from JCR journals

5.         The introduction resembles that of a review article and not that of a research article. What’s the gap of knowledge? Which is the scope of the manuscript? What hypothesis have been made? The introduction should be revised accordingly.

6.         Experimental section:. A more succinic yet complete writing should be done. Moreover the author state that a statistical analysis has been made. I believe that the authors should give more details about the analysis performed, 

7.         Results: Even though results are meant to be predominantly descriptive, it is helpful to give a bit of reasoning why a parameter was measured at the beginning of a section. As to the results, it may be clearer to describe results that are non-significant (and ns should be defined at first use) as the same, perhaps indicate a trend. Again it is really hard to understand which comparisons have been made. The statics conducted presence should be more accurately justified.  

8. Lettering are missing on the bars of charts so kindly add the lettering.

Fahad S, Hasanuzzaman M, Alam M, Ullah H, Saeed M, Ali Khan I, Adnan M. (Eds.) (2020) Environment, Climate, Plant and Vegetation Growth. Springer Nature Switzerland AG 2020. DOI: https://doi.org/10.1007/978-3-030-49732-3

Fahad, S., Sönmez, O., Saud, S., Wang, D., Wu, C., Adnan, M., Turan, V. (Eds.), 2021a. Plant growth regulators for climate-smart agriculture, First edition. ed, Footprints of climate variability on plant diversity. CRC Press, Boca Raton, FL.

Author Response

We would like to thank to the reviewer for the valuable contribution to manuscript. In this sense, we have carefully controlled and revised the manuscript, according to Reviewers’ comments.

Comment 1. English should improve by a native person. The paper suffers from a poor English structure throughout and cannot be published or reviewed properly in the current format. The manuscript requires a thorough proofread by a native person whose first language is English. The instances of the problem are numerous and this reviewer cannot individually mention them. It is the responsibility of the author(s) to present their work in an acceptable format. Unless the paper is in a reasonable format, it should not have been submitted.

Answer 1. Language checked and corrected by native speaker MSc. Ankur Kumar.

Comment 2. The novelty of the study needs to be highlighted compare to other similar studies.

Answer 2. The novelty of the research and its difference from other researches were tried to be emphasized in the introduction (Line 96-106).

Comment 3. Discussion is weak. The discussion needs enhancement with real explanations not only agreements and disagreements. Authors should improve it by the demonstration of biochemical/physiological causes of obtained results. Instead of just justifying results, results should be interpreted, explained to appropriately elaborate inferences. Discussion seems to be poor, didn't give good explanations of the results obtained. I think that it must be really improved. Where possible please discuss potential mechanisms behind your observations. You should also expand the links with prior publications in the area, but try to be careful to not over-reach. For the latter, you should highlight potential areas of future study.

Answer 3. Discussion was tried to improve according to reviewer comment.

Comment 4. The scientific background of the topic is poor. In "Introduction" and “Discussion", the authors should cite recent references between 2016-2020 from JCR journals.

Answer 4. Introduction and discussion were improved and the significant majority of references are articles published in the last 6 years.

Comment 5. The introduction resembles that of a review article and not that of a research article. What’s the gap of knowledge? Which is the scope of the manuscript? What hypothesis have been made? The introduction should be revised accordingly

Answer 5. The introduction revised accordingly.

Comment 6. Experimental section: A more succinic yet complete writing should be done. Moreover the author state that a statistical analysis has been made. I believe that the authors should give more details about the analysis performed,

Answer 6. The material and methods section revised accordingly.

Comment 7. Results: Even though results are meant to be predominantly descriptive, it is helpful to give a bit of reasoning why a parameter was measured at the beginning of a section. As to the results, it may be clearer to describe results that are non-significant (and ns should be defined at first use) as the same, perhaps indicate a trend. Again it is really hard to understand which comparisons have been made. The statics conducted presence should be more accurately justified. 

Answer 7. Physiological meanings of the parameters have been tried to be detailed in the manuscript where it thought to be missing. As the referee mentioned, since it is a study with too much data, statistically insignificant data were not mentioned in the manuscript unless it was considered important to be mentioned by the authors.

Comment 8. Lettering are missing on the bars of charts so kindly add the lettering.

Answer 8.   Lettering was added on the bars of charts and tables, excluding those calculated as percentage of control.

Reviewer 2 Report

-The paper compares two cultivars of safflower differing in their reaction under boron stress with NaCl pretreatment. Some results are interesting, however, there are many problems which should be solved.

1Line 35 Please mention boron threshold in plants

2 Line 39 please rewrite the following statement

3 Line 70-73 rewrite the following statement and I am unable to understand why the author used 75 mM NaCl treatment

How could the authors justify the choice of oilseed plant such as safflower in order to remedy boron toxicity

4 Line 46 Please do not start the sentences with abbreviation

5 The author needs to expand their introduction because the background knowledge given in introduction needs improvement

6 The authors needs to give clear hypothesis and its need

How did you select the two cultivars - screening of more cultivars?

7 Line 90-95 The language is bad, and the text is full of linguistic problems and typing errors, references as well.

8 Some statements from literature cited in Discussion does not correspond with the original texts, this is rather confusing.

9 Line 104 please rewrite the following statement

10 Line 114 please check the statement for grammar issue

11 These containers were place - please check English here and through the text

Some statements mentioned in the discussion do not correspond with the original text, very confusing, please check all your interpretation of other works

Author Response

We would like to thank to the reviewer for the valuable contribution to manuscript. In this sense, we have carefully controlled and revised the manuscript, according to Reviewers’ comments.

Comment 1. Line 35 Please mention boron threshold in plants

Answer 1. Boron threshold for plants was mentioned between Line 36-42.

Comment 2. Line 39 please rewrite the following statement.

Answer 2. The statement was rewritten.

Comment 3. Line 70-73 rewrite the following statement and I am unable to understand why the author used 75 mM NaCl treatment. How could the authors justify the choice of oilseed plant such as safflower in order to remedy boron toxicity

Answer 3. In Line 106-109, it was tried to explain how pretreatment concentration was determined. “The plants obtained from contaminated soil can be used industrially” phrase was added to conclusion.

Comment 4.  Line 46 Please do not start the sentences with abbreviation

Answer 4. The sentence revised accordingly.

Comment 5. The author needs to expand their introduction because the background knowledge given in introduction needs improvement

Answer 5. The introduction revised and expanded accordingly.

Comment 6. The authors needs to give clear hypothesis and its need How did you select the two cultivars - screening of more cultivars?

Answer 6. In Line 109-111, it was tried to explain that cultivars were selected from our previous studies.

Comment 7. Line 90-95 The language is bad, and the text is full of linguistic problems and typing errors, references as well.

Answer 7. The entire manuscript revised accordingly.

Comment 8. Some statements from literature cited in Discussion does not correspond with the original texts, this is rather confusing.

Answer 8. Since boron stress is associated with abiotic stress responses, especially drought, references thought to be confusing were used in the literature. All those references have been changed upon the referee's valuable opinion. Some articles that is not particularly related to boron toxicity, have been used for general expressions describing the parameter.

Comment 9. Line 104 please rewrite the following statement

Answer 9. The statement rewritten.

Comment 10. Line 114 please check the statement for grammar issue

Answer 10. The sentence revised.

Comment 11. These containers were place - please check English here and through the text

Answer 11. The entire manuscript was edited by native speaker.

Comment 12. Some statements mentioned in the discussion do not correspond with the original text, very confusing, please check all your interpretation of other works

Answer 12. All discussion section and references revised accordingly.

Reviewer 3 Report

The research paper is interest and well written

The study contains valuble information about salt sress in Safluer

The paper must be improved by revision the typos and some details to the material and methods should be added

Author Response

We would like to thank to the reviewer for the valuable contribution to manuscript. In this sense, we have carefully controlled and revised the manuscript, according to Reviewers’ comments.

Comment 1. The paper must be improved by revision the typos and some details to the material and methods should be added

Answer 1. The entire manuscript was improved. The material and methods section revised and detailed accordingly.

Reviewer 4 Report

Please indicate in the page 1 line 36 the difference of boric acid and borate.

In page 15 the title of Scheme 1 pleas rearrange.

Author Response

We would like to thank to the reviewer for the valuable contribution to manuscript. In this sense, we have carefully controlled and revised the manuscript, according to Reviewers’ comments.

Comment 1. Please indicate in the page 1 line 36 the difference of boric acid and borate.

Answer 1. The difference of Boric acid and Borate was tried to explain in Line 42-47.

Comment 2. In page 15 the title of Scheme 1 pleas rearrange.

Answer 2. Scheme 1 title was rewritten and detailed.

Round 2

Reviewer 2 Report

Yes, I am satisfied with revision, so I recommend acceptation of the article for "Plants" journal.

Author Response

We are grateful to Reviewer for reviewer’s contribution for the manuscript.